# Penicillides from *Penicillium* and *Talaromyces*: Chemical Structures, Occurrence and Bioactivities

**DOI:** 10.3390/molecules29163888

**Published:** 2024-08-16

**Authors:** Maria Michela Salvatore, Rosario Nicoletti, Filomena Fiorito, Anna Andolfi

**Affiliations:** 1Department of Chemical Sciences, University of Naples ‘Federico II’, 80126 Naples, Italy; mariamichela.salvatore@unina.it (M.M.S.); anna.andolfi@unina.it (A.A.); 2Council for Agricultural Research and Economics, Research Centre for Olive, Fruit and Citrus Crops, 81100 Caserta, Italy; 3Department of Agricultural Sciences, University of Naples ‘Federico II’, 80055 Naples, Italy; 4Department of Veterinary Medicine and Animal Production, University of Naples ‘Federico II’, 80137 Naples, Italy; filomena.fiorito@unina.it; 5BAT Center-Interuniversity Center for Studies on Bioinspired Agro-Environmental Technology, University of Naples ‘Federico II’, 80138 Naples, Italy

**Keywords:** secondary metabolites, Eurotiomycetes, fungal extrolites, depsidones, bioactivity

## Abstract

Penicillide is the founder product of a class of natural products of fungal origin. Although this compound and its analogues have been identified from taxonomically heterogeneous fungi, they are most frequently and typically reported from the species of *Talaromyces* and *Penicillium*. The producing strains have been isolated in various ecological contexts, with a notable proportion of endophytes. The occurrence of penicillides in these plant associates may be indicative of a possible role in defensive mutualism based on their bioactive properties, which are also reviewed in this paper. The interesting finding of penicillides in fruits and seeds of *Phyllanthus emblica* is introductory to a new ground of investigation in view of assessing whether they are produced by the plant directly or as a result of the biosynthetic capacities of some endophytic associates.

## 1. Introduction

Fungi in the genus *Penicillium* (Eurotiomycetes, Aspergillaceae) represent one of the most exploited sources of chemodiversity, with a multitude of structural models and compound classes having been reported since the discovery of mycophenolic acid by Gosio [1]. Many species of these fungi have been described from all ecological contexts and geographical areas in the world, which are able to produce blockbuster drugs such as penicillin and compactin [2]. However, the recent affirmation of the principle ‘one fungus, one name’ in taxonomy [3] and the widespread employment of biomolecular markers for a more accurate identification have brought the separation of *Penicillium* species having symmetrical biverticillate conidiophores into the genus *Talaromyces* (Eurotiomycetes, Trichocomaceae) based on phylogenetic reconstructions [4].

Fifty years ago, a Japanese researcher reported on the finding of a new secondary metabolite from an isolate of *Penicillium* sp., which was named penicillide (11-hydroxy-3-[(1*S*)-1-hydroxy-3-methylbutyl]-4-methoxy-9-methyl-5H,7H-dibenzo[b,g][1,5]dioxocin-5-one) (Figure 1) [5]. This product is the founder of a class of bioactive products characterized by 2,4-dihydroxybenzilic alcohol and 2-hydroxy-4-methoxy-benzoic acid moieties linked together by ether and ester bonds, forming a typical 8-membered heterocycle in the place of the 7-membered one usually found in depsidones [6]. Rather than being one of a kind, more analogues of penicillide were later found by other independent research groups in the early 1990s; possibly due to difficulties in accessing the pertinent literature at that time, some of these compounds were given the unrelated names of purpactins [6] and vermixocins [7].

This paper offers an overview on the research activity which has been developed in the last five decades on penicillides, with reference to their occurrence in nature as secondary metabolites of *Penicillium* and *Talaromyces* species, as well as to their biological properties and possible biotechnological applications.

## 2. Chemical Structures

With reference to the general molecular scaffold outlined above, the A ring is invariable in all the known penicillides with the exception of hydroxypenicillide, while the C ring presents a side chain of five carbons at C-3. Substitutions in this chain vary among the ten analogues which have been identified from these fungi so far (Table 1). Besides searching databases, such as PubMed, Google Scholar and Web of Science, an accurate analysis of the available data based on the chemical structure was carried out using Scifinder.

With reference to the structures in Table 1, it must be noted that isopenicillide (**9**) is not an isomeric form of penicillide (**1**) since it has an additional hydroxyl group on the side chain in ring C; rather, it is an isomer of 5′-hydroxypenicillide (**7**). Moreover, the denomination adopted for compounds **7** and **8** can be misleading, and it is incoherent because compound **8** has an additional hydroxyl group on the A ring.

## 3. Occurrence

Conforming to the above-introduced separation of symmetrically biverticillate species in *Talaromyces*, most of the strains producing penicillides are found to belong to this genus (Table 2). Similar to *Penicillium*, the latter taxon has been characterized as an outstanding source of chemodiversity [8,9,10,11], exhibiting some peculiar biosynthetic models which have been only or predominantly found in these fungi, such as the funicones [12,13].

In light of the above, it is quite possible that the isolates provisionally identified as *Penicillium* sp. actually belong to *Talaromyces*. This can be directly verified for strain MA-37 through a blast in GenBank of the reported rDNA-ITS sequence [17], while the doubt remains in the other cases. However, apart from these incompletely identified strains, the fact that penicillides have been reported in three unrelated species, namely *P. chrysogenum P. montanense* and *P. simplicissimum*, represents an indication that these compounds may be common secondary metabolites in *Penicillium*, too. Yet, the connotation of penicillides as products characterizing the biosynthetic abilities of *Talaromyces* is well supported by the remark that as many as 18 species of this genus are listed in Table 1 as documented sources.

More infrequent are reports of these products from the taxonomically related genera *Aspergillus* (Aspergillaceae) and *Neosartorya* (Trichocomaceae), that is, penicillide [55,56,57], purpactin A, dehydropenicillide and Δ^2′^-1′-dehydropenicillide [58], while other occasional findings concern miscellaneous Ascomycetes species. This is the case of *Alternaria* (Dothideomycetes, Pleosporaceae), reported to produce penicillide [59]; *Guignardia* (= *Phyllosticta*: Dothideomycetes, Botryosphaeriaceae), producing purpactin A and prenpenicillide (identified as (*E*)-3-(3-methylbut-1-enyl)-11-hydroxy-4-methoxy-9-methyl-7H-dibenzo[b,g][1,5]dioxocin-5-one) [60]; *Scytalidium cuboideum* (Leotiomycetes, *incertae sedis*), producing purpactin A [61]; *Colpoma quercinum* (Leotiomycetes, Rhytismataceae), producing penicillide [62]; and *Pestalotiopsis* spp. (Sordariomycetes, Pestalotiopsidaceae), producing penicillide and purpactin A [63], dehydroisopenicillide and 3′-O-methyldehydroisopenicillide [64]. Fungi in the genus *Pestalotiopsis* have also been found to produce pestalotiollides A-B and sinopestalotiollides A-D; these are structural analogues with a modified side chain, which have not been reported from *Talaromyces* and *Penicillium* so far [64,65].

The assumption that penicillides are typical fungal products has been impaired by an intriguing report on the extraction in equable amounts of penicillide, purpactin A and the novel analogue 1′*S*-11-dehydroxypenicillide from fruits of the Indian gooseberry (*Phyllanthus emblica*) (Malpighiales, Phyllanthaceae) [66]. This finding has been followed by the detection of penicillide in seeds of the same plant by an independent research group [67]. Indeed, recent progresses in natural product research have disclosed the ability of taxonomically diverse endophytic fungi to synthesize compounds originally characterized from plants [68], which, in many cases, has postulated the transfer of gene clusters encoding for their synthesis [69,70]. By extension of this concept, horizontal gene transfer (HGT) could also have operated in the case of *P. emblica*, but in which direction? Did the plant acquire the genetic base from any endophytic fungus, or rather, did it occur that several endophytes borrowed this gene cluster from the plant and later spread to the various ecological contexts from which the penicillide producers have been reported? Alternatively, the extraction from *P. emblica* could be consequential to its production and accumulation in the plant tissues as resulting from the biosynthetic capacities of one or more endophytic associates, as has been demonstrated in the case of defensive mutualists of ryegrass and other plants [71]. Indeed, both *Talaromyces* and *Penicillium* species are at the forefront among endophytic fungi, having been reported from many and diverse plants in all environments [11,72], and the hypothesis that a systematic association established with *P. emblica* could lead to the accumulation of penicillides in its fruits and seeds deserves to be verified. The very recent report of the species *Talaromyces atroroseus* and *Penicillium choerospondiatis* from gooseberries [73] opens a new ground of investigation in this respect.

Finally, while this manuscript was in preparation, another paper was published reporting on the detection of purpactin A in the aqueous extract obtained from the Chinese vine *Sargentodoxa cuneata* (Ranunculales, Lardizabalaceae) [74]; undoubtedly, this last finding reinforces the need to more thoroughly examine plants as possible sources of penicillides.

## 4. Biological Properties

Biological properties have been essentially evaluated for the two most common products, penicillide and purpactin A; the available data are shown in Table 3.

It is generally agreed that the bioactivities of microbial secondary metabolites are related to their competitive interactions in the biocenosis. In this respect, both penicillide and purpactin A exhibit moderate antibacterial properties, which anyway could have an ecological impact considering the possible synergism with other antimicrobial compounds produced by *Penicillium* and *Talaromyces* species. Although various strain panels have been used in the antibacterial assays, the same MICs for both products generally resulted when they were concomitantly tested (e.g., against *Klebsiella pneumoniae*, *Pseudomonas aeruginosa* and *Vibrio parahaemolyticus*) [12]. The highest activities were detected for penicillide at 0.78 µg mL^−1^ against the methicillin and oxacillin resistant strain ATCC 43300 of *Staphylococcus aureus* [27], while purpactin A was active at a 5-fold higher concentration (4 µg mL^−1^) against *E. coli* [26]. In the latter study, a positive correlation was observed between the acetylation of the side chain and antibacterial efficacy [26].

Antifungal activity was evaluated for penicillide against the opportunistic pathogenic Basidiomycete *Cryptococcus neoformans*, but effectiveness only resulted at a quite high concentration [20]. High active concentration also resulted in assays of this compound against brine shrimps [4], while it displayed antifouling properties against *Balanus amphitrite* at a lower concentration than purpactin A [34]. Conversely, antiplasmodial effects were observed at lower concentrations for purpactin A [20].

Assays for cytotoxic activity yielded quite variable results. In fact, on Hep G2 (hepatocellular carcinoma) cells, cytotoxicity was higher for penicillide, as assessed in two different laboratories [6,15], while incongruent values were obtained in two other laboratories for purpactin A on MCF-7 (breast cancer) cells [17,20]. In assays carried out at the same laboratory, it was slightly higher for penicillide against KB (epidermoid carcinoma) cells and slightly higher for purpactin A against Vero (kidney epithelium of African green monkey) cells [20]. Finally, the same values were measured for the two compounds in terms of incorporation of uridine, thymidine and valine in P388 (murine leukemia) cells [25].

Limited information has been gathered with reference to possible biotechnological applications related to miscellaneous enzyme inhibitory activities. These properties were approximately in the same range against α-glucosidase [33,39], while cholesterol acyltransferase inhibition was detected at lower concentrations for purpactin A, as measured in rabbit liver microsomes [8]. Anticholesterolemic activity of the latter compound was also documented in rat liver microsomes and J774 macrophages [36,41]. Moderate activity was detected for penicillide as a calpain inhibitor with possible applications for the treatment of muscular dystrophy and neurodegenerative diseases [2]; moreover, the reported activity of penicillide as an oxytocin antagonist [26] could be exploited in gynecology. Finally, purpactin A was found to consistently act as an elastase inhibitor with possible application in the treatment of chronic obstructive pulmonary disease [42] and as an inhibitor of TMEM16A, a Ca^2+^-activated Cl^−^ channel protein involved in mucus secretion in inflamed airways, which has been proposed as a drug target for diseases associated with mucus hypersecretion, including asthma. The compound prevented Ca^2+^-induced mucin release in cytokine-treated airway cells, while it did not affect cell viability, epithelial barrier integrity and activities of membrane transport proteins essential for maintaining airway hydration [77].

## 5. Biosynthesis of Penicillide and Purpactin A

Penicillides are structurally related compounds biosynthetically derived from the polyketide pathway. It has been proposed that the skeleton of these natural products, characterized by 2,4-dihydroxybenzilic alcohol acid and 2,4-dihydroxybenzoic acid moieties linked by ether and ester bonds, derives from chrysophanol anthrone after the decarboxylation of a single octaketide chain. The subsequent oxidative cleavage of the B ring of chrysophanol anthrone is assumed to generate a benzophenone intermediate which is then oxidized to produce the tricyclic skeleton (spirobenzofuran-l,2′-cyclohexa-3′,5′-diene-2′,3-dione). Further structural modifications (i.e., prenylation, acetylation and methylation) of this latter compound generate purpactin B which is synthesized as an intermediate; then, the oxidation of its hydroxymethyl group leads to purpactin A [78,79].

The biosynthesis of penicillide can be deduced starting from the intermediate spirobenzofuran-l,2′-cyclohexa-3′,5′-diene-2′,3-dione (Figure 2). Slight modifications of the side chain or the addition of functional groups on the tricyclic skeleton (dibenzo[b,g][1,5]dioxocin-5(7H)-one) may generate the other known compounds of this class.

## 6. Related Products

Several products of *Penicillium* and *Talaromyces* are structurally related to penicillides, but they cannot be strictly considered members of this class of natural products because of the lack of the typical structural features (Figure 3). This is the case of purpactins B and C which, despite having been named as purpactin A homologues, are characterized by spirobenzofuran-l,2′-cyclohexa-3′,5′-diene-2′,3-dione instead of dibenzo[b,g][1,5]dioxocin-5(7H)-one. Considering the biosynthetic pathway in Figure 2, purpactin B is an intermediate in the biosynthesis of purpactin A, which is formed by the acetylation and rearrangement of the former compound. The common biosynthetic origin of purpactins could explain why these compounds were frequently reported from the same fungal sources [6,37,52].

It has been proposed that secopenicillides A, B and C could share the same biosynthetic route of penicillide and purpactin A [46]. However, these compounds, and the related 7-*O*-acetylsecopenicillide C [17], are characterized by the presence of a diphenylether moiety which does not conform to the genuine structural model of penicillides. In particular, Wu and coworkers proposed that secopenicillide A originates from purpactin C which is converted to purpactin A via reduction of the aldehydic group to form an alcohol intermediate, followed by esterification [46]. Secopenicillides A and B were isolated for the first time from *P. simplicissimum* [23], while secopenicillide C was detected as a new compound from *T. pinophilus,* and its production was found to be enhanced in co-culture with *Trichoderma harzianum* [45]. It must be noted that all these metabolites were detected along with other known penicillides.

## 7. Conclusions

As can be concluded from the available information on their occurrence and structures examined in this review, species of *Penicillium* and *Talaromyces* from various ecological contexts represent the main natural source of penicillides. However, the reported extraction of these depsidones from two unrelated plant species introduces the opportunity for further investigations in the aim to assess whether, or not, this biosynthetic aptitude results from any endophytic associates. In the latter alternative, further studies are also advisable to investigate if the inter-organism spread derives from the transfer of a pertinent gene cluster through HGT, which may eventually confer an ecological advantage depending on the bioactive properties of these compounds.

The broad-range laboratory investigations carried out so far have pointed out the multifaceted bioactivities of penicillides. Particularly, the best antibiotic effects have been documented against methicillin and oxacillin resistant *S. aureus* [27], deserving further assessments in the perspective to integrate the available panel of these highly demanded drugs. In this respect, a relevant contribution is expected by the pharmaceutical industry through the realization of more potent semi-synthetic derivatives, as a follow-up of the pioneering research activity carried out in some laboratories [80,81]. Considering the increasing impact of pulmonary diseases, a notable contribution in the achievement of new drugs is also expected from further investigations on the above-outlined effects on mucus hypersecretion [42]. Finally, following the preliminary evidence obtained for depsidones [82,83], more valuable opportunities could result from the exploration of the antiviral effects of these compounds.

## Figures and Tables

**Figure 1 molecules-29-03888-f001:**
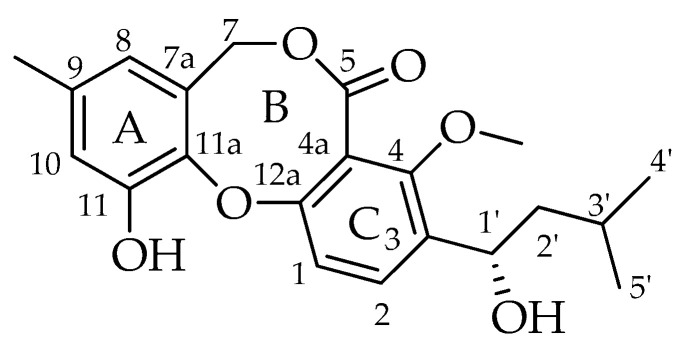
Structure of penicillide (**1**).

**Figure 2 molecules-29-03888-f002:**
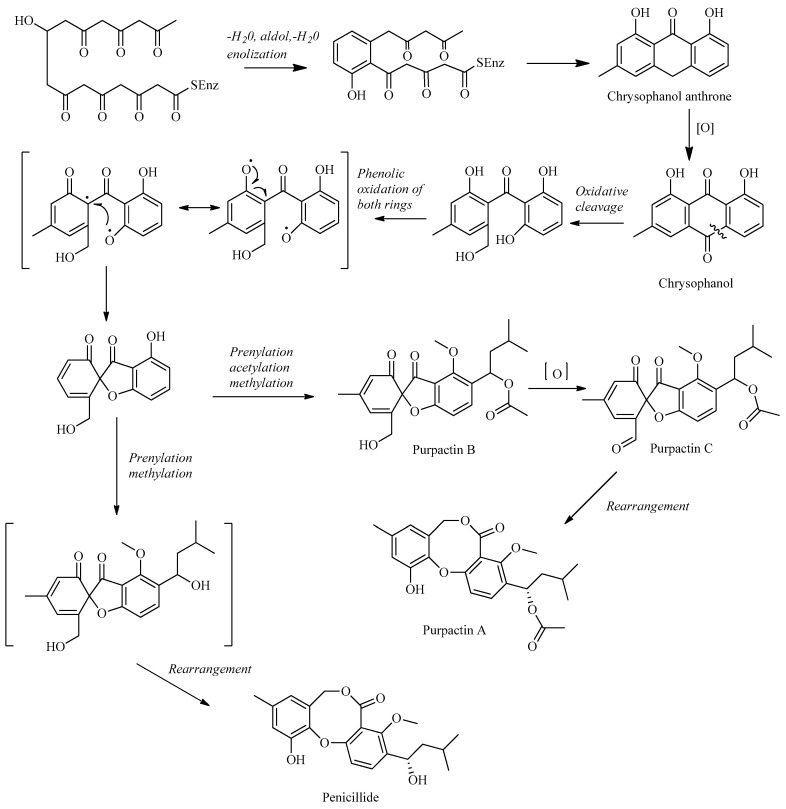
The proposed biosynthetic schemes of penicillide and purpactin A.

**Figure 3 molecules-29-03888-f003:**
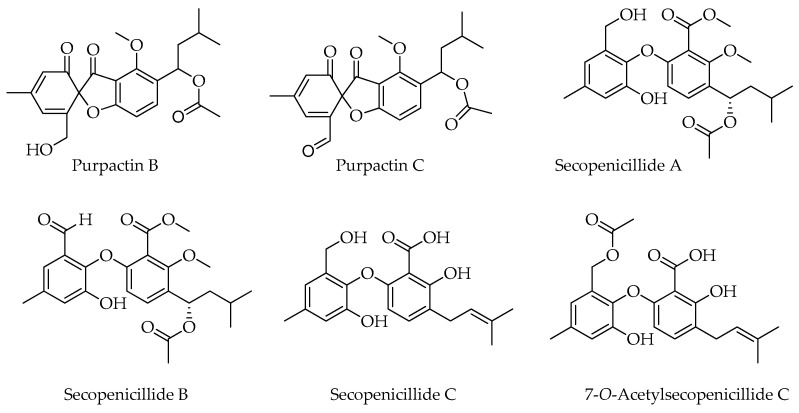
Compounds structurally and/or biosynthetically related to penicillides reported from *Penicillium* and *Talaromyces*.

**Table 1 molecules-29-03888-t001:** Penicillides from *Panicillium* and *Talaromyces*.

Code	Compounds	Chemical Structure	Nominal Mass	Formula
**1**	Penicillide (= vermixocin A)	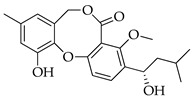	372	C_21_H_24_O_6_
**2**	Purpactin A (= vermixocin B)	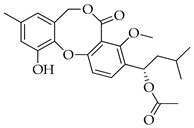	414	C_23_H_26_O_7_
**3**	Δ^1′,3′^-1-Dehydroxypenicillide	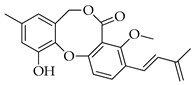	352	C_21_H_20_O_5_
**4**	1′2′-Epoxy-3′,4′-didehydropenicillide	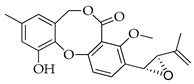	368	C_21_H_20_O_6_
**5**	Dehydroisopenicillide (= MC-141, 1,2-dehydropenicillide)	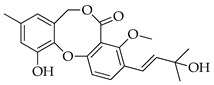	370	C_21_H_22_O_6_
**6**	2′-Hydroxy-3′,4′-didehydropenicillide	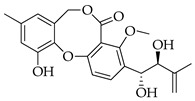	386	C_21_H_22_O_7_
**7**	5′-Hydroxypenicillide	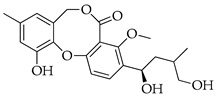	388	C_21_H_24_O_7_
**8**	Hydroxypenicillide	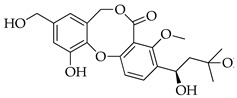	404	C_21_H_24_O_8_
**9**	Isopenicillide	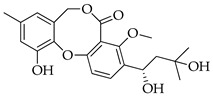	388	C_21_H_24_O_7_
**10**	3′-*O*-Methyldehydroisopenicillide (= 3-methoxy-1′2-dehydropenicillide)	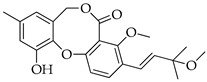	384	C_22_H_24_O_6_
**11**	Prenpenicillide	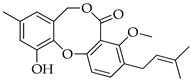	354	C_21_H_22_O_5_

**Table 2 molecules-29-03888-t002:** *Penicillium* and *Talaromyces* species/strains reported as producers of penicillides.

Species	Substrate	Location	Compounds	Ref.
*Penicillium* sp.	-	Japan	**1**	[5]
*Penicillium* sp.	endophytic in *Taxus cuspidata*	Gunma (Japan)	**4**, **6**, **5**, **10**	[14]
*Penicillium* sp. F60760	soil	South Korea	**1**	[15]
*Penicillium* sp. H1	marine	China	**1**	[16]
*Penicillium* sp. MA-37 ^1^	rhizosphere of *Bruguiera gymnorrhiza*	Hainan (China)	**1**, **3**, **5, 10**	[17]
*Penicillium* sp. PF9	endophytic in *Artemisia princeps*	South Korea	**1**, **10**	[18]
*Penicillium* sp. SCS-KFD16	marine	China	**1**	[19]
*Penicillium* sp. ZLN29	marine sediment	Jiazhou Bay (China)	**1**, **11**	[20]
*P. chrysogenum* MT-12	endophytic in *Huperzia serrata*	China	**1**, **2**, **8**, **9**	[21]
*P. montanense* KYI635 ^2^	soil	Kanagawa (Japan)	**1**, **2**	[22]
*P. simplicissimum* IFM53375	unspecified	Japan	**1**, **2**	[23]
*Talaromyces* sp.	moss (*Climacium dendroides*)	Gangneung (South Korea)	**1**, **2**	[24]
*Talaromyces* sp. BTBU20213036	mud from intertidal zone	Qingdao (China)	**1**	[25]
*Talaromyces* sp. CS-258	mussel from cold seep	South China Sea	**1**, **2**, **5**	[26]
*Talaromyces* sp. HK1-18	mangrove soil	Hainan (China)	**1**	[27]
*Talaromyces* sp. IQ-567	endophytic in *Rhizophora mangle*	Tecomate Lagoon (Mexico)	**1**	[28]
*Talaromyces* sp. LF458	sponge (*Axinella verrucosa*)	Elba island (Italy)	**3**, **10**	[29]
*Talaromyces* sp. M27416	sea water	Dongshan island (China)	**1**	[30]
*Talaromyces* sp. Wangcy005	gorgonian (*Subergorgia suberosa*)	Weizhou island (China)	**1**, **2**, **3**	[31]
*Talaromyces* sp. WHUF0362	mangrove soil	Hainan (China)	**2**	[32]
*T. aculeatus* IBT23209	soil	Araracuara (Colombia)	**1**, **2**	[33]
*T. aculeatus* IBT23210			**2**	
*T. aculeatus* IBT23211			**2**	
*T. aculeatus* IBT23212			**2**	
*T. aculeatus* PSU-RSPG105	soil	Rajjaprabha Dam (Thailand)	**1**, **2**	[34]
*T. albobiverticillius* IBT4466	imported pomegranate	Denmark	**2**	[35]
*T. albobiverticillius* CBS113167	air in cake factory	-	**2**	
*T. amestolkiae* CBS433.62	ground domestic waste	Verona (Italy)	**2**	[36]
*T. amestolkiae* CBS436.62	alum solution	-	**2**	
*T. atroroseus* IBT3933	-	-	**2**	[35]
*T. atroroseus* TRP-NRC	agricultural waste	Egypt	**2**	[37]
*T. derxii* NHL2981	soil	Kurashiki (Japan)	**1**, **5**	[38]
*T. flavus* CCMF-276	soil	Jàchymov (Czechia)	**1**, **2**	[39]
*T. flavus* ATCC74110	-	-	**1**	[40]
*T. flavus* TGGP34	endophytic in *Acanthus ilicifolius*	China	**1**, **2**, **5**	[41]
*T. fuscoviridis* CBS193.69	soil	the Netherlands	**2**	[42]
*T. gwangjuensis* CNUFC-WT19-1	freshwater	Yeosu (South Korea)	**2**	[43]
*T. koreana* CNUFC-YJW2-13	freshwater	Yeosu (South Korea)	**2**	
*T. pinophilus* AF-02	endophytic in *Allium fistulosum*	Lanzhou (China)	**1**	[44]
*T. pinophilus* FKI-5653	soil	Hachijo island (Japan)	**1**, **5**	[45]
*T. pinophilus* H608	mangrove sediment	Xiamen (China)	**1**, **2**, **3**, **5**, **7**, **9**	[46]
*T. pinophilus* LT6	tobacco rhizosphere	Lecce area (Italy)	**1**	[47]
*T. pinophilus* SCAU037	rhizosphere of *Rhizophora stylosa*	Techeng isle (China)	**1**, **9**, **10**	[48]
*T. pinophilus* XS-20090E18	unidentified gorgonian	Xisha islands (China)	**1**, **2**, **8**, **9**	[49]
*T. purgamentorum* CBS113145	forest leaf litter	Peña Roja (Colombia)	**1**, **2**	[33]
*T. purpureogenus* CFRM02	unspecified	Karnataka (India)	**1**	[50]
*T. purpureogenus* FO-608	soil	Japan	**2**	[6]
*T. purpureogenus* ATCC44445	corn kernel	Georgia (USA)	**2**	[36]
*T. purpureogenus* ATCC20204	-	Japan	**2**	
*T. purpureogenus* CBS286.36	-	Japan	**2**	
*T. purpureogenus* IBT17540	barley	Winnipeg (Canada)	**2**	
*T. purpureogenus* IBT11632	imported marjoram	Denmark	**2**	
*T. purpureogenus* IBT12779	marine	France	**2**	
*T. purpureogenus* IMI112715	rhizosphere of *Trifolium alexandrinum*	Egypt	**2**	
*T. purpureogenus* IMI136126	molded corn	Wisconsin (USA)	**2**	
*T. purpureogenus* IMI136127	molded corn	Wisconsin (USA)	**2**	
*T. purpureogenus* NRRL3290	-	Georgia (USA)	**2**	
*T. purpureogenus* MM	cotton textile	Egypt	**1**, **10**	[51]
*T. ruber* CBS237.93	unknown	Unknown	**2**	[36]
*T. ruber* CBS113160	-	-	**2**	
*T. ruber* CBS132699	sandy soil	Sousse (Tunisia)	**2**	
*T. ruber* FRR1503	preserved wood	Australia	**2**	
*T. ruber* IBT31167	-	-	**2**	
*T. stellenboschiensis* CBS135665	soil	Stellenbosch (South Africa)	**2**	[42]
*T. stipitatus* SK-4	leaf of *Acanthus ilicifolius*	Guangxi (China)	**1**, **2**	[52]
*T. thailandensis* PSU-SPSF059	soil	Thailand	**2**	[53]
*T. veerkampii* CBS500.78	soil	Meta (Colombia)	**1**, **2**	[42]
*T. veerkampii* CBS136668	soybean seed	Matou (Taiwan)	**1**, **2**	[42]
*T. verruculosus* TGM14	mangrove (*Xylocarpus granatum*)	Hainan (China)	**1**	[54]

^1^ This strain should be more correctly ascribed to *Talaromyces* based on an updated GenBank blast of its deposited ITS sequence. ^2^ This strain was originally identified as *Penicillium asperosporum*.

**Table 3 molecules-29-03888-t003:** Biological activities of penicillide and purpactin A.

Biological Activity	Concentration	Results and Further Details	Ref.
**Penicillide (1)**			
Antibacterial	100 µg mL^−1^	*Acinetobacter baumannii* (40% inhibition)	[28]
	50 µg mL^−1^	*Clostridium perfringens* (MIC)	[44]
	64 µg mL^−1^	*Escherichia coli* (MIC)	[26]
	64 µg mL^−1^	*Klebsiella pneumoniae* (MIC)	[26]
	50 µg mL^−1^	*Micrococcus tetragenus* (MIC)	[44]
	32 µg mL^−1^	*Pseudomonas aeruginosa* (MIC)	[26]
	100 µg mL^−1^	*Staphylococcus aureus* (MIC)	[25]
	0.78 µg mL^−1^	MRSA *S. aureus* (MIC)	[27]
	64 µg mL^−1^	*Vibrio alginolyticus* (MIC)	[26]
	32 µg mL^−1^	*Vibrio parahaemolyticus* (MIC)	[26]
Antifouling	2.6 µg mL^−1^	*Balanus amphitrite* (EC_50_)	[49]
Antifungal	128 µg mL^−1^	*Cryptococcus neoformans* (MIC)	[34]
Anti-inflammatory	11.5 µM	RAW264.7 (IC_50_)	[18]
Antimalarial	16.41 µM	*Plasmodium falciparum* (IC_50_)	[34]
Brine shrimp lethal	158.5 µM	LD_50_	[17]
Cholesterol acyltransferase inhibition	22.9 µM	rabbit liver microsomes (IC_50_)	[22]
Cytotoxic	50 µg mL^−1^	P388	[39]
	9.7 µM	Hep G2 (IC_50_)	[20]
	6.7 µM	HEp-2 (IC_50_)	[49]
	43.77 µM	KB (IC_50_)	[34]
	7.8 µM	RD (IC_50_)	[49]
	53.73 µM	Vero (IC_50_)	[34]
	50 µg mL^−1^	incorporation of uridine, thymidine and valine in P388	[39]
m-Calpain inhibition	7.1 µM	SLLVY-AMC (IC_50_)	[15]
Oxytocin binding inhibition	67 µM	IC_50_	[40]
α-Glucosidase inhibition	78.4 µM	IC_50_	[48]
**Purpactin A (2)**			
Antibacterial	8 µg mL^−1^	*Aeromonas hydrophila* (MIC)	[26]
	4 µg mL^−1^	*E. coli* (MIC)	
	2.42 µmol L^−1^	*Helicobacter pylori* 129 (MIC)	[32]
	4.83 µmol L^−1^	*H. pylori* G27 (MIC)	
	64 µg mL^−1^	*K. pneumoniae* (MIC)	[26]
	64 µg mL^−1^	MRSA *S. aureus* (MIC)	
	8 µg mL^−1^	*Micrococcus luteus* (MIC)	
	32 µg mL^−1^	*P. aeruginosa* (MIC)	
	16 µg mL^−1^	*Vibrio anguillarum* (MIC)	
	8 µg mL^−1^	*Vibrio harveyi* (MIC)	
	32 µg mL^−1^	*V. parahaemolyticus* (MIC)	
Antifouling	4.8 µg mL^−1^	*B. amphitrite* (IC_50_)	[31]
	10 µg mL^−1^	*B. amphitrite* (EC_50_)	[49]
Antimalarial	5.69 µM	*P. falciparum* (IC_50_)	[34]
Cholesterol acyltransferase inhibition	120 µM	rat microsomes (IC_50_)	[6,75]
	1.2 µM	J774 (IC_50_)	
	8.2 µM	rabbit liver microsomes (IC_50_)	[22]
Cytotoxic	15.1 µM	HCT-116 (IC_50_)	[31]
	38.95 µM	Hep G2 (IC_50_)	[29]
	50 µg mL^−1^	incorporation of uridine, thymidine and valine in P388	[39]
	9.7 µM	J774 (IC_50_)	[6,75]
	52.5 µM	KB (IC_50_)	[34]
	16.4 µM	MCF-7 (IC_50_)	[31]
	75.28 µM	MCF-7 (IC_50_)	[34]
	41.21 µM	NIH 3 T3 (IC_50_)	[29]
	32.57 µM	Vero (IC_50_)	[34]
Elastase inhibition	37.2 µg mL^−1^	IC_50_	[76]
*α*-Glucosidase inhibition	80.9 µM	IC_50_	[52]

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
