# Peer review of "Penicillides from Penicillium and Talaromyces: Chemical Structures, Occurrence and Bioactivities"

_molecules, 2024, doi:10.3390/molecules29163888_

Round 1

Reviewer 1 Report

Comments and Suggestions for Authors

This manuscript reviews the Penicillides from Penicillium and Talaromyces: Chemical Structures, Occurrence and Bioactivities. Overall, the content of this manuscript is relatively simple. Only 11 Penicillide related compounds were reported, the number of compounds is very limited. There is also a lack of very attractive content in biological properties and biosynthesis.

Author Response

Thank you for reviewing our manuscript.

Reviewer 2 Report

Comments and Suggestions for Authors

Many thanks to the authors for their efforts in preparing this review, which I see as written in a proper system and arrangement, but I have some comments;

- The abstract must be reformulated to clarify the basic hypothesis of the research, especially the biological activities and their importance

- The introduction and conclusion also does not mention the biological activities and their importance

-  The discussion must be refined by deeply discussing the biological activities and the mechanism of action of each one, even in brief

- Based on what was reported in this review, what are your future vision and recommendations?

- In table 2: The data attributed to reference No. 33 does not belong to him, please check it (It is cited twice in the same table).

- In Table No. 3, some abbreviations, such as Hep G2, KB, and RD, are the first time they are mentioned, their synonyms must be written in the table (in brief ).

Reference list:

-Line 238: correct the reference No to be “2” instead of “1”

-Line 270: write “Artemisia princeps” in italic

-Line 323: write “T. verruculosus” in italic

-Line 330: write “Trichoderma harzianum” in italic

The attached file could be helpful

Author Response

Many thanks to the authors for their efforts in preparing this review, which I see as written in a proper system and arrangement, but I have some comments;

Thank you for your positive comments.

- The abstract must be reformulated to clarify the basic hypothesis of the research, especially the biological activities and their importance

- The introduction and conclusion also does not mention the biological activities and their importance

-  The discussion must be refined by deeply discussing the biological activities and the mechanism of action of each one, even in brief

Actually, there is no discussion section in this review. However, there is a proper section (4. Biological properties) in which the main biological activities of penicillide and purpactin A were discussed. Furthermore, some considerations on this subject have been integrated in the conclusion section.

- Based on what was reported in this review, what are your future vision and recommendations?

We have added further considerations on these issues following the reviewer’s suggestions.

- In table 2: The data attributed to reference No. 33 does not belong to him, please check it (It is cited twice in the same table).

We apologize for this oversight. Actually, reference 33 is another paper by the same first author, which has been replaced in the updated version.

- In Table No. 3, some abbreviations, such as Hep G2, KB, and RD, are the first time they are mentioned, their synonyms must be written in the table (in brief ).

Indeed, these are not abbreviations but codes associated to the cell lines which do not require any explanation.

Reference list:

-Line 238: correct the reference No to be “2” instead of “1”

-Line 270: write “Artemisia princeps” in italic

-Line 323: write “T. verruculosus” in italic

-Line 330: write “Trichoderma harzianum” in italic

All the above corrections have been done.

Reviewer 3 Report

Comments and Suggestions for Authors

This manuscript entitled “Penicillides from Penicillium and Talaromyces: Chemical Structures, Occurrence and Bioactivities” by  Salvatore, et al provides a comprehensive overview of penicillides, a class of natural products found in fungi belonging to the genera Penicillium and Talaromyces. The authors effectively cover the following aspects:

Chemical structures: The core structure and variations among different penicillide analogues are well-described.

·  Occurrence: The primary source of penicillides is identified as Penicillium and Talaromyces species, with interesting secondary findings from other fungi and even plants.

·  Biological properties: The review summarizes the reported antibacterial, antifungal, antiplasmodial, cytotoxic, and enzyme inhibitory activities of penicillides.

·  Biosynthesis: A proposed pathway for penicillide biosynthesis is presented.

·  Related products: The manuscript clarifies the distinction between penicillides and structurally related compounds like purpactins B and C and secopenicillides.

The review provides a thorough analysis of penicillides, encompassing their chemistry,

occurrence, bioactivities, and biosynthesis. The manuscript is structured logically, making it easy to follow the discussion. Introduction is too lengthy and should be shortened.

The conclusions part is very poor, should be elaborated, discussion on future research avenues for penicillides would be beneficial.

Including figures depicting chemical structures and the proposed biosynthetic pathway would enhance clarity.

The authors propose horizontal gene transfer (HGT) as a possible explanation for the presence of penicillides in plants. Can you elaborate on the evidence supporting this hypothesis, or are there alternative explanations that could be explored?

The manuscript mentions the potential application of penicillides in various therapeutic areas. Can you discuss the specific challenges that need to be addressed before penicillides can be developed into viable drugs?

Given the structural similarity between penicillides and their analogues, can we expect similar biological activities from these related compounds? Are there any reports on the bioactivity of purpactins B and C or secopenicillides?

Author Response

This manuscript entitled “Penicillides from Penicillium and Talaromyces: Chemical Structures, Occurrence and Bioactivities” by  Salvatore, et al provides a comprehensive overview of penicillides, a class of natural products found in fungi belonging to the genera Penicillium and Talaromyces. The authors effectively cover the following aspects:

Chemical structures: The core structure and variations among different penicillide analogues are well-described.

  • Occurrence:The primary source of penicillides is identified as Penicillium and Talaromyces species, with interesting secondary findings from other fungi and even plants.
  • Biological properties:The review summarizes the reported antibacterial, antifungal, antiplasmodial, cytotoxic, and enzyme inhibitory activities of penicillides.
  • Biosynthesis:A proposed pathway for penicillide biosynthesis is presented.
  • Related products:The manuscript clarifies the distinction between penicillides and structurally related compounds like purpactins B and C and secopenicillides.

The review provides a thorough analysis of penicillides, encompassing their chemistry, occurrence, bioactivities, and biosynthesis. The manuscript is structured logically, making it easy to follow the discussion. 

Thank you for your positive comments.

Introduction is too lengthy and should be shortened.

Indeed, the introduction section is limited to essential information; we just cannot shorten this text.

The conclusions part is very poor, should be elaborated, discussion on future research avenues for penicillides would be beneficial.

Conclusions have been further elaborated.

Including figures depicting chemical structures and the proposed biosynthetic pathway would enhance clarity.

According to the referee's comment, the figure and text concerning the biosynthetic pathway of penicillide and purpactin A have been modified.

The authors propose horizontal gene transfer (HGT) as a possible explanation for the presence of penicillides in plants. Can you elaborate on the evidence supporting this hypothesis, or are there alternative explanations that could be explored?

There is no experimental evidence supporting the HGT hypothesis. However, considering the literature recently published in the field (cfr. references 68-70), HGT as the biological phenomenon explaining the biosynthetic affinities between plants and endophytes is more than a mere conjecture.

The manuscript mentions the potential application of penicillides in various therapeutic areas. Can you discuss the specific challenges that need to be addressed before penicillides can be developed into viable drugs?

As required, further discussion concerning these aspects has been added in the conclusions section.

Given the structural similarity between penicillides and their analogues, can we expect similar biological activities from these related compounds? Are there any reports on the bioactivity of purpactins B and C or secopenicillides?

We pointed out that the latter compounds do not belong to the penicillides; hence, their bioactivity was not considered.

Reviewer 4 Report

Comments and Suggestions for Authors

The present review paper deals with a particular class of compounds viz. penicillides from Penicillium and Talaromyces. Specifically, their chemical structures, the occurrence and their bioactivities are covered. In such a context, the paper offers an overview on the research activity developed in the last five decades on the penicillides, as secondary metabolites of Penicillium and Talaromyces species, as well as to their biological properties and possible biotechnological applications.

The work is interesting since starting from 1974, when a Japanese researcher reported on the finding of a new secondary metabolite from an isolate of Penicillium sp., many  analogues of penicillide were later found by other independent research groups in the early 1990s. As a matter of fact, taking into consideration also the issues in accessing the proper literature at that time, some of these compounds were given the unrelated names of purpactins and vermixocin.

Overall, the review is well-structured and in most of its parts well-written. However, I am very concerned about the similarity of the present review paper with the one published by the same research group in 2016 and was neglected by the authors (Mar. Drugs 2016, 14, 37) where the bioactive compounds produced by strains of Penicillium and Talaromyces of marine origin were reported. In fact, the introduction is too maigre and does not report a critical comparison with other reviews on the same topic. Thus, the novel aspects of the present review over the other published ones must be emphasized. Further, there is no reference to the search strategy employed for literature retrieval, including databases searched, keywords used, and inclusion/exclusion criteria.

Table 3. For consistency, the concentrations of penicillide and purpactin A should be reported with the same unit.

Comments on the Quality of English Language

Moderate revision

Author Response

The present review paper deals with a particular class of compounds viz. penicillides from Penicillium and Talaromyces. Specifically, their chemical structures, the occurrence and their bioactivities are covered. In such a context, the paper offers an overview on the research activity developed in the last five decades on the penicillides, as secondary metabolites of Penicillium and Talaromyces species, as well as to their biological properties and possible biotechnological applications.

The work is interesting since starting from 1974, when a Japanese researcher reported on the finding of a new secondary metabolite from an isolate of Penicillium sp., many  analogues of penicillide were later found by other independent research groups in the early 1990s. As a matter of fact, taking into consideration also the issues in accessing the proper literature at that time, some of these compounds were given the unrelated names of purpactins and vermixocin.

Overall, the review is well-structured and in most of its parts well-written.

Thank you for your positive comments.

However, I am very concerned about the similarity of the present review paper with the one published by the same research group in 2016 and was neglected by the authors (Mar. Drugs 2016, 14, 37) where the bioactive compounds produced by strains of Penicillium and Talaromyces of marine origin were reported.

We disagree that this manuscript is similar to the review which was published by R.N. along with Antonio Trincone. Indeed, that one considered ALL secondary metabolites produced by Penicillium and Talaromyces of MARINE ORIGIN only, while this paper refers to a SINGLE CLASS of compounds produced by strains of ANY ORIGIN. Moreover, more than 8 years have been passed since then, which remarkably increased data on occurrence and properties of this compound family. The previous review was not mentioned as a reference to avoid over-citation of our previous work.

In fact, the introduction is too maigre and does not report a critical comparison with other reviews on the same topic. Thus, the novel aspects of the present review over the other published ones must be emphasized. Further, there is no reference to the search strategy employed for literature retrieval, including databases searched, keywords used, and inclusion/exclusion criteria.

Further discussion concerning these aspects has been added in the conclusions section, and the strategy employed for literature retrieval has been described in the section “2. Chemical Structures”.

Table 3. For consistency, the concentrations of penicillide and purpactin A should be reported with the same unit.

We prefer to leave the units used in each original paper to avoid any misinterpretation of the data. 

Round 2

Reviewer 3 Report

Comments and Suggestions for Authors

the manuscrispt was imprived and ready for acceptance

Reviewer 4 Report

Comments and Suggestions for Authors

The authors did a great job in addressing all Reviewer's remarks. The manuscript can be accepted for publication.